# Does Route of Full Feeding Affect Outcome among Ventilated Critically Ill COVID-19 Patients: A Prospective Observational Study

**DOI:** 10.3390/nu14010153

**Published:** 2021-12-29

**Authors:** Dimitrios Karayiannis, Sotirios Kakavas, Aikaterini Sarri, Vassiliki Giannopoulou, Christina Liakopoulou, Edison Jahaj, Aggeliki Kanavou, Thodoris Pitsolis, Sotirios Malachias, George Adamos, Athina Mantelou, Avra Almperti, Konstantina Morogianni, Olga Kampouropoulou, Anastasia Kotanidou, Zafeiria Mastora

**Affiliations:** 1Department of Clinical Nutrition, “Evangelismos” General Hospital of Athens, Ypsilantou 45-47, 10676 Athens, Greece; aalmperti@yahoo.gr (A.A.); konstantina.morogianni@gmail.com (K.M.); 2Intensive Care Unit, Center for Respiratory Failure, “Sotiria” General Hospital of Chest Diseases, 152 Mesogeion Avenue, 11527 Athens, Greece; sotikaka@yahoo.com; 3First Department of Critical Care Medicine and Pulmonary Services, Evangelismos General Hospital, National and Kapodistrian University of Athens, 11527 Athens, Greece; katsarri5@hotmail.com (A.S.); vaso.giannop88@gmail.com (V.G.); chr1stieliak@gmail.com (C.L.); edison.jahaj@gmail.com (E.J.); agg_kan@hotmail.com (A.K.); theodorepitsolis@yahoo.com (T.P.); sotmalachias@gmail.com (S.M.); George.adamos1983@gmail.com (G.A.); athina.mantelou@gmail.com (A.M.); olgakampou@yahoo.com (O.K.); akotanid@gmail.com (A.K.); zafimast@yahoo.gr (Z.M.)

**Keywords:** SARS-CoV-2 virus, energy target, enteral nutrition, parenteral nutrition, critical illness

## Abstract

The outbreak of the new coronavirus strain SARS-CoV-2 (COVID-19) highlighted the need for appropriate feeding practices among critically ill patients admitted to the intensive care unit (ICU). This study aimed to describe feeding practices of intubated COVID-19 patients during their second week of hospitalization in the First Department of Critical Care Medicine, Evaggelismos General Hospital, and evaluate potential associations with all cause 30-day mortality, length of hospital stay, and duration of mechanical ventilation. We enrolled adult intubated COVID-19 patients admitted to the ICU between September 2020 and July 2021 and prospectively monitored until their hospital discharge. Of the 162 patients analyzed (52.8% men, 51.6% overweight/obese, mean age 63.2 ± 11.9 years), 27.2% of patients used parenteral nutrition, while the rest were fed enterally. By 30 days, 34.2% of the patients in the parenteral group had died compared to 32.7% of the patients in the enteral group (relative risk (RR) for the group receiving enteral nutrition = 0.97, 95% confidence interval = 0.88–1.06, *p* = 0.120). Those in the enteral group demonstrated a lower duration of hospital stay (RR = 0.91, 95% CI = 0.85-0.97, *p* = 0.036) as well as mechanical ventilation support (RR = 0.94, 95% CI = 0.89–0.99, *p* = 0.043). Enteral feeding during second week of ICU hospitalization may be associated with a shorter duration of hospitalization and stay in mechanical ventilation support among critically ill intubated patients with COVID-19.

## 1. Introduction

The recent COVID-19 pandemic, which resulted from SARS-CoV-2 coronavirus infection, contributed to a rapid increment in hospital and intensive care unit (ICU) admissions [1]. Most patients exhibit mild to moderate symptoms, such as fever, cough, fatigue, respiratory failure, and multiple effects on the gastrointestinal tract, which appear especially in patients who are over 60 years old and those with concomitant diseases [2]. Approximately 5% of patients develop severe disease and require ICU admission, where the provision of adequate nutritional support is a challenge given the complex and fluctuating metabolic changes that accompany their nutritional status over time [3]. Critical illness, in general, is characterized by an inflammatory response that elicits a catabolic response and can be divided into different metabolic diseases: the early acute phase, which occurs in the first 48 h after entering the ICU, and the next, called the late acute phase, which usually lasts days after admission [4]. In severely ill patients with COVID-19, the acute phase presents with a prolonged severe inflammatory response, which is associated with a significant increase in resting energy expenditure [5,6], and current strategies for nutritional therapy seem to be unsuccessful in covering energy/protein needs [7]. In addition, COVID-19 patients spend extended periods in the ICU compared to the usual severely ill [8], and thus, metabolic phases of the disease are likely to be different and last longer.

Nutritional support is essential for all critically ill COVID-19 patients because acute illness carries an increased risk of malnutrition, muscle wasting, and increased mortality and requires a complex calculation of feeding route, time of onset, and amount and type of nutrients [4,9]. All the above may significantly affect patient’s outcomes, such as length of hospital stay or duration of mechanical ventilation [10]. Recently, various guidelines from international societies were developed and published early in the pandemic without providing convincing answers to many critical questions [3,11,12]. One important question is route of nutritional support administration (enteral vs. parenteral), the effects of which on outcome remain unclear [13,14]. Up to the present moment, administration of energy needs via the enteral route is the preferred method of choice according to published guidelines [15,16] although it is affiliated with higher rates of gastrointestinal intolerance [17]. Parenteral feeding is more invasive, provides higher amounts of calories, and is associated with a higher risk of complications [18]. There are several published papers comparing different types of interventions regarding the route of feeding and ICU outcome but mainly among non COVID-19 patients during their first days of ICU stay [19].

Despite it being quoted in various guidelines, there is lack of data comparing enteral vs. parenteral feeding among critically ill patients with COVID-19 disease as well as the effect of full feeding (after 5–7 days of ICU stay) on clinical outcomes [20,21,22,23]. Therefore, the purpose of this study was to describe full feeding practices and highlight their long-term consequences among intubated patients with COVID-19 disease, who were treated within ICU at a reference hospital in Athens.

## 2. Methods

### 2.1. Participants

We conducted this single-center observational study from September 2020 to July 2021 in the first intensive care unit of Evaggelismos Athens General Hospital (tertiary hospital). Adult patients (aged > 18 years), requiring mechanical ventilation for >48 h, projected to receive nutritional support for at least 5 days, and with laboratory-confirmed COVID-19 by a positive reverse-transcriptase polymerase-chain-reaction (RT-PCR) assay of a nasopharyngeal swab specimen were eligible. The Institutional Review Board at Athens Evangelismos Hospital: 116/31-03-2021 approved data collection and waived the need for informed consent. Exclusion criteria were existence of metabolic diseases requiring a specific diet (for example phenylketonuria), history of gastrectomy or esophagectomy, initiation of nutrition support 5–7 days before ICU admission, ICU length of stay (LOS) shorter than 7 days, pregnancy, or patients that were expected to require no more than 48 h of invasive mechanical ventilation.

### 2.2. Study Design

Data were retrieved from patient electronic file and daily hospital sheet on the admission, 7th day, 14th day, and until discharge from hospital. During ICU hospitalization, anthropometric data, such as height (in cm) and weight (in kg), were collected, while body mass index (in kg/m^2^) was determined. Besides patients’ basic characteristics, the Acute Physiology and Chronic Health Evaluation (Apache) score was calculated 24 h after ICU admission. After the initiation of nutrition support, various data, such as pre-existing illness, initiation/duration of mechanical ventilation, resource to organ support (vasopressors and kidney replacement therapy), use of prone position, and laboratory markers, such as serum albumin and CRP levels, were also computed.

### 2.3. Feeding Practices

Nutritional support was initiated as soon as possible after admission to the ICU and no later than 48 h after intubation. Data on feeding practices and tolerance to feeding were collected every day by the Nutrition Support Team (NST), while nutrient delivery was based on ESPEN guidelines by the hospital NST [4]. Feeding practices included enteral and parenteral feeding, while nutritional parameters assessed included calorie (from nutrient and non-nutrient calorie sources such as propofol and dextrose) and protein intake Regarding caloric requirements, calculations were based on previously published indirect calorimetry measurements in a subset of patients [6]. In our ICU, we had established protocols for nutrition support from the first day of the COVID-19 pandemic, based on the ESPEN guidelines or local consensus. The NST made the decision on the nutrition regimen as well as the route of nutrition support in collaboration with the treating physician.

Patients in the enteral nutrition group were fed through a nasogastric or nasoduodenal catheter depending on tolerance and comorbidities. The aim was to meet the caloric requirements by the 7th day of hospitalization using isosmotic, isocaloric, high-protein, polymeric preparations, after which the decision was at the discretion of the bedside physician [9]. In all cases, nutrition support was administered continuously (mL per h). Malnourished patients (BMI < 17 or clinical diagnosis) were assigned to a parenteral nutrition protocol designed to reduce the risk of refeeding syndrome [11,16]. In the parenteral group (presence of uncontrolled shock, uncontrolled life-threatening hypoxemia, hypercapnia or acidosis, hemodynamic instability, or when energy intake was consistently <50% of targets over 5–7 days), all patients received only parenteral nutrition via a central venous catheter depending on the results of daily haemodynamic assessments and nonfunctional digestive tract. We excluded from the analysis patients using supplemental parenteral nutrition to achieve predefined calorie targets after day 8. Parenteral nutrition was stopped and replaced with enteral nutrition at the flow rate needed to achieve the pre-defined calorie target once the patient met pre-defined criteria for haemodynamic stability. Bedside physician provided extra water, electrolytes, vitamins, and trace elements using standard preparations.

### 2.4. Outcome

The primary outcome was death by day 30. Secondary outcomes included mortality up to day 60 of hospitalization, ICU mortality, length of hospital and ICU stay, days of mechanical ventilation support, days of renal replacement, and percentage of patients who received mechanical ventilation, vasopressors and renal replacement therapy. 

### 2.5. Statistical Analysis

The categorical variables are expressed as frequencies while the continuous variables as means (with standard deviations) or medians (with interquartile ranges (IQRs)), according to the rejection/not rejection of null hypothesis by the Shapiro–Wilk test, and then compared using Students *t*-test or Mann–Whitney U test. Due to the lack of information regarding effect of the feeding route on COVID-19 patients at the time of study design, it was not possible to perform a sample size calculation prior to the beginning of the current study when we performed hypothesis testing. The comparison between the groups was made using the Wilcoxon rank sum tests for continuous variables, while a chi-square test of independence (***χ***^2^) (Fisher exact tests where appropriate) was performed for categorical variables. We used generalized linear models to test associations between feeding groups (enteral vs. parenteral nutrition) and study final outcomes. A Poisson distribution with log-link function was used to test association of number of count data, while a Binomial distribution with logit link function was used for clinical endpoints. The results are presented as relative risk (RR) and 95% confidence intervals (CIs). Statistical tests were considered significant if *p* < 0.05. All statistical analyzes were conducted using SPSS version 21.0.

## 3. Results

### 3.1. Patients

From September 2020 to July 2021, 192 patients with a confirmed SARS-CoV-2 infection during the second and third wave of the COVID-19 pandemic were admitted to the Evangelismos intensive care unit (ICU). Of these, 16 patients were excluded from the study because they either died within the first 48 h after admission or were under nutritional support, while four had mechanical ventilation started more than 24 h earlier, resulting in 176 patients (Figure 1). Participants were intubated within 24 h of admission, and their mean age was 62.1 ± 10.9 years, while 51.2% were male. The demographic, baseline, and nutritional characteristics of the patients analyzed are presented in Table 1. There was no clear categorization of patients as malnourished or not at admission, as it was not possible to collect information on dietary intake, weight loss, and the usual body weight. Almost half of the participants had Nutric Score values >5, indicating a high nutritional risk upon admission to the ICU.

### 3.2. Outcomes

Data on feeding practices are presented in Table 1. A total of 117 individuals were fed enterally, while the rest were fed through the parenteral route. Of those belonging to the parenteral nutrition group, 5.6% experienced side effects, such as electrolyte disturbances, pneumothorax, hypoglycemia, and cholestasis, while in those receiving enteral nutrition, vomiting (32.3%) was the most common side effect.

Fifteen of 45 patients (33.3%) in the parenteral group and 38 of 117 patients (34.2%) in the enteral group died, with no significant between-group difference by day 30 even after adjusting for various baseline factors, such as age, sex, and race, APACHE score, Nutric Score (as a dichotomous variable), BMI, diabetes, and chronic kidney disease (relative risk in the enteral group, 0.97; 95% confidence interval (CI), 0.88 to 1.06, Table 2). There were significant reductions in the enteral group as compared with the parenteral group for in-hospital length of stay (RR = 0.92; 95% CI, 0.86 to 0.98; *p* = 0.039), ventilator days (RR = 0.94; 95% CI, 0.89 to 0.99; *p* = 0.043), and elevated liver enzymes ((RR = 0.91; 95% CI, 0.85 to 0.97; *p* = 0.039); *p* = 0.022). However, there were also significant differences regarding the rates of adverse gastrointestinal events between the parenteral group and the enteral group (32.3% vs. 22.1% for vomiting and 37.2% vs. 29.2% for diarrhea, *p* < 0.05). There was no significant difference in the duration of survival up to 60 days. Initiation of feeding was delayed in 37 patients in the parenteral nutrition group and in 41 patients in the enteral nutrition group, while caloric and protein intake are shown in Table 1. The energy target of 25–30 calories per kilogram of body weight per day was reached in the majority of patients in both groups, and the average caloric intake was almost identical in both groups.

## 4. Discussion

Τhis single-center prospective study evaluated the effect of the nutritional support administration route after the seventh day of hospitalization on outcome among critically ill intubated COVID-19 patients. Our data suggest that enteral feeding is superior to parenteral feeding in terms of length of hospital stay and mechanical ventilation, but these findings were not accompanied by a corresponding improvement in mortality both in-hospital and out-of-hospital at 60 days. These two groups also did not exhibit any differences regarding infections incidence, days on RRT, and septic shock prevalence.

Are there other data available to date on feeding practices and their effect on outcome among COVID-19 critically ill patients? Only a few studies so far have provided limited data about nutrition support. A USA cohort [24] revealed that more than half of the participants (56%) presented intolerance to enteral nutrition, which was associated with higher ICU stay and in-hospital mortality, whereas a similar study among intubated patients from Mexico revealed a lower prevalence of intolerance to enteral nutrition—about 32% [25]. A series of 176 critically ill patients with COVID-19 disease [26] managed to reach their energy and protein requirements during the first week of admission especially through the use of supplemental parenteral nutrition. Parenteral nutrition use was comparable to our study, as approximately 35% of patients were fed through the parenteral route compared with 27.7% of our patients. Reports from another study—the ISIIC point prevalence study—also suggest that there is a growing interest in the role of nutrition support, which resulted in providing COVID-19 patients with higher amounts of energy and protein compared to non-COVID-19 patients [27].

There is also lack of available data regarding the effect of feeding administration after the first week of ICU hospitalization when energy and protein targets are theoretically achieved [26]. All previous major studies tried to investigate the effect of either early administration (during the first 72 h following admission) of nutritional support [28] either up to the fifth day of hospitalization [19,29] or the first week of hospitalization [14] on outcome. The two largest multicenter clinical trials among critically ill patients assessing the effects of nutritional support route during first week of ICU stay suggest nonsignificant difference in all-cause mortality, frequency of infectious complications, ICU, and hospital length of stay (LOS) [14,19]. In our cohort, feeding practices were investigated during the second week of hospitalization and not the first. This was settled for two reasons: so far, most studies that explore the relationship between feeding practices during the first week of ICU and outcome have not provided any clear association [10,14,19,21]. The other reason is that in critically ill patients, energy metabolic demands usually peak within five to seven days before returning to normal (ebb/flow phases) [30]. Regarding COVID-19 disease, there is a stable hypermetabolic condition probably because of hyperinflammatory response, which persists during the second week of hospitalization and appears to stabilize after the 10th day of ICU stay, as recent studies have demonstrated [5,6].

The total length of hospital stay was lower in those patients receiving enteral nutrition although ICU length of stay (LOS) was not different. A systematic review and meta-analysis show that early enteral feeding is associated with a reduced postoperative length of stay among patients undergoing lower gastrointestinal surgery [31], while among critically ill patients, route of nutrition support during first week of hospitalization was not related to LOS although there was a trend for reduced ICU stay [14]. Harvey et al. revealed that there was no difference in both length of hospital and ICU stay among critically ill patients according to feeding rout e [19]. Another main finding of this study was that those receiving enteral feeding presented a shorter duration of mechanical ventilation support. As compared to earlier times, enteral feeding is no longer contraindicated in patients under mechanical ventilation and is the preferred route of feeding administration [32], while some data do not suggest that enteral feeding is superior [33]. On the other hand, parenteral feeding is well recognized for its higher caloric intake, leading to hyperglycemia and increased ventilator stay [34].

In addition to these findings, parenteral nutrition group was more likely to experience an increase in liver enzymes during hospitalization. Research on the effects of parenteral nutrition has focused on the possibility of liver dysfunction as well as an increase in liver enzymes [35]. This phenomenon is a potential result of cholestasis, which is caused by biliary obstruction or impaired secretion of bile [36]. On appearance, γ-glutamyl transpeptidase, alkaline phosphatase, and conjugated bilirubin levels are elevated. This is a clinical manifestation accompanying long-term parenteral nutrition therapy among pediatric patients or adults. Long-term PN therapy-induced cholestasis is a significant consequence that can progress to cirrhosis and liver failure [37].

The present study was not without limitations. First, our study had a relatively small sample size, which limits the generalizability to other ICUs and countries. Second, the prospective nature of this study does not define a causal relationship and reflects only associations between the implemented nutrition protocols and study outcomes. Moreover, we did not obtain data regarding energy expenditure by using indirect calorimetry in the whole sample because it was very time consuming to get such a large quantity of data. Our study was not a randomized evaluation of feeding practices; even if the indication for parenteral and enteral nutrition followed guidelines and local treatment protocols, a selection bias cannot be excluded. We should also point out that a higher proportion of patients belonging to the parenteral nutrition group exhibited high Nutric Score values, but we have controlled for this confounding factor in the multivariate models. Furthermore, it was not possible to assess all possible covariates among our patients, and future studies might lead to different results regarding the provision of nutrition therapy in critically ill patients.

## 5. Conclusions

There are scarce data on feeding practices among critically ill COVID-19 patients and their effects on outcome. Our study demonstrated that the majority of COVID-19 intubated patients were fed through the enteral route during their second week of ICU hospitalization. Enteral nutrition was not superior compared to parenteral feeding in relation to the main study outcome, which was 30-day mortality, but it was associated with reduced length of hospital stay, less demand for mechanical ventilation support, and a more favorable profile for liver enzyme levels.

## Figures and Tables

**Figure 1 nutrients-14-00153-f001:**
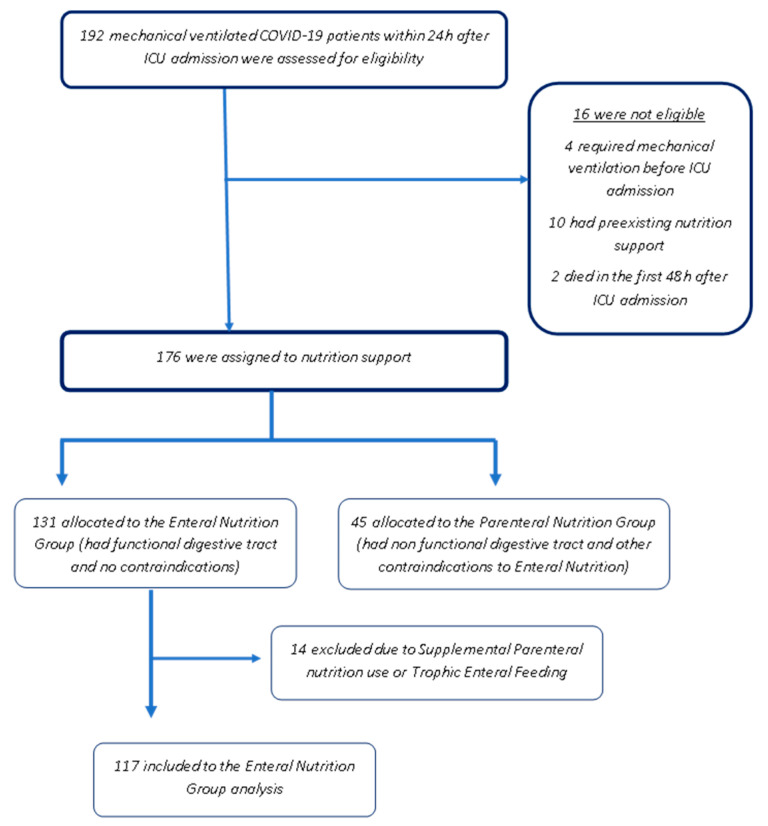
Patient recruitment flow chart.

**Table 1 nutrients-14-00153-t001:** Baseline characteristics of the participants (*n* = 162).

Characteristics	Parenteral Nutrition Group *n* = 45 (27.7%)	Enteral Nutrition Group *n* = 117 (72.3%)	*p*-Value
Age (year)	62.7 ± 10.7	63.2 ± 11.9	0.181
Sex, male (*n*) %	21 (46.6)	62 (52.9)	0.093
Active Smoker (*n*)%	9 (20.0)	21 (17.8)	0.233
Comorbidities, (*n*) %			0.112
Hypertension	23(51.1)	62 (52.9)	
Diabetes	21 (46.6)	52 (44.4)	
COPD	7 (15.5)	16 (13.6)	
Chronic Renal Failure	2 (4.4)	3 (2.5)	
Nutritional data			
BMI on admission, (kg/m^2^), (*n*) %			0.233
Normal (18.5–24.9)	16 (36.2)	45 (38.5)	
Overweight (25–29.9)	12 (26.6)	31 (26.4)	
Obese (≥30)	14 (31.1)	41 (35.0)	
NUTRIC Score on admission, (*n*) %			0.046 *
Low risk (0–4 points)	21 (46.7)	15 (59.7)	
High risk (5–9 points)	24 (53.3)	57 (40.3)	
Fluid balance (mL/day)	1250 ± 215	1015 ± 188	
Coverage of energy need during 7–14th day of ICU stay (*n*) %	89.1	86.5	0.076
Protein delivered during ICU (g/kg ideal body weight/day)	1.09 ± 0.61	1.17 ± 0.68	0.122
Time from ICU admission to start nutrition (IQR)—h	17.8 (13.4–27.2)	22.3 (15.2–31.2)	0.041 *******
Calories administered—kcal/kg of body weight/day	27.8 ± 7.8	26.3 ± 6.9	0.098
Clinical Data			
APACHE II score on admission	18.3 ± 6.8	17.1 ± 6.2	0.249
Use of prone positioning, *n* (%)	26 (58.4)	67 (57.2)	0.135
PaO_2_/FiO_2_ ratio (mmHg)	126 (92-170)	128 (96–171)	0.338
Serum albumin g/L	3.16 ± 0.80	3.05 ± 0.96	0.224
Vasopressor therapy, *n* (%)	33 (73.3)	83 (70.9)	0.336
Side effects, *n* (%)			0.039 *******
Electrolyte disturbances	2 (4.4)	1 (0.8%)	
Vomiting	10 (22.1)	37 (31.6%)	
Diarrhea	13 (29.2)	43 (36.7%)	
Hypoglycemia	-	-	
Other (cholestasis, pneumothorax)	1 (1.2)	-	

Values represent median (IQR) or means (+SD) or number of subjects (*n*, %). * Denotes statistically significant different between groups at <0.05 level, *p* = *p* value for Students *t*-test or Mann-Whitney *U* test or Chi square test. Abbreviations: APACHE, Acute Physiology and Chronic Health Evaluation; COPD, chronic obstructive pulmonary disease; BMI, body mass index; PEEP, positive end expiratory pressure; Nutric Score, Nutrition Risk in the Critically Ill; Fi02, fraction of inspired oxygen; NMBAs, neuromuscular blocking agents.

**Table 2 nutrients-14-00153-t002:** Patients primary and secondary outcomes (*n* = 162).

Outcome	Parenteral Nutrition Group (*n* = 45)	Enteral Nutrition Group (*n* = 117)	Relative Risk (95% CI) ^#^	*p*-Value
Primary				
Death within 30 days, *n* (%)	15 (33.3)	38 (32.4)	0.97 (0.88–1.06)	0.120
Secondary				
Death, *n* (%)				
In-hospital mortality,	14 (31.1%)	36(30.7%)	0.98 (0.86–1.10)	0.132
ICU mortality,	17 (37.9%)	43 (36.7%)	0.96 (0.85–1.08)	0.124
60-day mortality	16 (35.5%)	41 (35%)	0.97 (0.82–1.10)	0.233
Hospital length of stay (days)	35 (7–59)	30 (8–52)	0.92 (0.86–0.98)	0.039 *
ICU length of stay (days)	23 (6–51)	21 (7–49)	0.98 (0.90–1.06)	0.078
Ventilator days (30-day study period only)	21 (6–28)	17 (6–24)	0.94 (0.89–0.99)	0.043 *
Days on RRT (30-day study period only)	17 (5–28)	18 (6–29)	0.98 (0.89–1.07)	0.180
Kidney failure requiring RRT	13 (28.8%)	34 (29.1%)	0.95 (0.81–1.09)	0.337
Tracheostomy, *n* (%)	11 (24.4%)	29 (24.7%)	0.96 (0.83–1.09)	0.197
ICU acquired Infections, *n* (%)	7 (15.5%)	20 (17.1%)	0.89(0.72–1.06)	0.221
Septic shock, *n* (%)	30 (66.6%)	75 (64.1%)	0.94 (0.86–1.02)	0.063
Elevated liver enzymes	13 (28.8%)	17 (14.5%)	0.91 (0.85–0.97)	0.022 *

Values represent median (IQR) or means (+SD) or number of subjects (*n*, %). * Denotes statistically significant different between groups at <0.05 level **^#^** Covariates were selected a priori, incorporating demographic information (age, sex, and race). APACHE score, Nutric Score (dichotomous variable), BMI, diabetes, and chronic kidney disease. Abbreviations: ICU, intensive care unit; RRT, renal replacement therapy.

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
