# Peer review of "Does Route of Full Feeding Affect Outcome among Ventilated Critically Ill COVID-19 Patients: A Prospective Observational Study"

_nutrients, 2021, doi:10.3390/nu14010153_

Round 1

Reviewer 1 Report

Thank you very much for this interesting and relevant article.

Here are some minor remarks regarding the lay out.

TEXT

Line 109

“ICU unit” has to be changes into “ICU”, since the “U” stands for “unit”.

Line 177

With regard to content, the chapter “3.2 Feeding practices” belongs to “chapter 3.3 outcome”. Please incorporate.

Line 180-182

Please incorporate these important findings in the tables as well.

Line 240

The abbreviation “LOS” needs to be introduced when you use it the first time. Please check if all other abbreviations are introduced correctly.

Line 292

Stay in columns when giving the abbreviations and writing them out.

TABLES

Both tables need revision regarding the lay out.

  • Use the same system continuously, giving units. Write in every line either “n (%)” or “mean (SD) or “median (IQR) or whatever is applicable.

  • Lines 164 and 172

Add to both titles whereof the percentage is calculated.

  • Mark as headlines:

Comorbidities, BMI on admission

  • Use the same kind of typing for all headlines.

  • Write “Nutrition” always with capital N.

  • Mark every significant p-value with “*”.

  • What do you mean with “IBW” in “ICU (g/kg IBW/day)”?

Please correct typing errors

  • BMI on admission, (kgr/m2) à BMI on admission (kg/m2)
  • NUTRIC Score at admission, n (%) à NUTRIC Score at admission n (%)
  • Erase the symbol after “body weight per day”

Author Response

We would like to thank Reviewers #1 for the comments on our manuscript. In the next lines, a point-by-point response to each comment is presented.

Thank you very much for this interesting and relevant article.

Here are some minor remarks regarding the lay out.

Line 109 “ICU unit” has to be changes into “ICU”, since the “U” stands for “unit”.

We thank Reviewer#1 for the helpful comments. As suggested, the word “unit” was deleted (lines 164 in the revised MS).

Line 177 With regard to content, the chapter “3.2 Feeding practices” belongs to “chapter 3.3 outcome”. Please incorporate.

We thank Reviewer#1 for this comment. As suggested, feeding practices chapter was incorporated to 3.2 chapter, entitled outcomes  (line 228 in the revised MS).

Line 180-182 Please incorporate these important findings in the tables as well.

 We thank Reviewer#1 for this comment. As suggested, these findings were inserted on table 2 (lines 499 in the revised MS).

Line 240 The abbreviation “LOS” needs to be introduced when you use it the first time. Please check if all other abbreviations are introduced correctly.

  We thank Reviewer#1 for this comment. As suggested, all abbreviations were introduced before their first appearance in text.

Line 292 Stay in columns when giving the abbreviations and writing them out.

   We thank Reviewer#1 for the comment. As suggested, all abbreviations were now corrected.

TABLES

 Both tables need revision regarding the lay out.

Use the same system continuously, giving units. Write in every line either “n (%)” or “mean (SD) or “median (IQR) or whatever is applicable.

  We thank Reviewer#1 for this comment. As suggested, all tables were revised according to suggested comments.

Lines 164 and 172 Add to both titles where of the percentage is calculated.

   We thank Reviewer#1 for this comment. We have now added the percentage.  

Mark as headlines: Comorbidities, BMI on admission

  • Use the same kind of typing for all headlines.
  • Write “Nutrition” always with capital N.
  • Mark every significant p-value with “*”.
  • What do you mean with “IBW” in “ICU (g/kg IBW/day)”?

  We thank Reviewer#1 for this comment. As suggested, all tables were revised according to suggested comments.

Please correct typing errors

  • BMI on admission, (kgr/m2) à BMI on admission (kg/m2)
  • NUTRIC Score at admission, n (%) à NUTRIC Score at admission n (%)
  • Erase the symbol after “body weight per day”

  We thank Reviewer#1 for this series of comments. As suggested, all errors were corrected according to suggested comments.

Reviewer 2 Report

The article describes feeding practices in critically ill patients admitted to an ICU due to a severe SARS CoV-2 infection. After careful examination, it is clear that the manuscript requires significant alternations before it is ready for publishing in the journal. The manuscript suffers from numerous flaws, including many significant issues such as:

General comments:

Both grammar and syntax must be improved - proper language editing is obligatory

Introduction:

The authors failed to provide a convincing rationale for performing another study on the nutritional intervention in COVID-19. Furthermore, the introduction section is overloaded with insignificant data, primarily speculative. Please add information on the recently published guidelines (ESPEN, SSC, ASPEN) and rewrite the aims of the study, which are not clearly defined. 

Methodology

The manuscript would benefit from adding the STROBE checklist

The inclusion criteria are not described in sufficient detail - please use the STROBE checklist for guidance. 

The feeding practices section requires significant modification:

  • According to the recent guidelines collection of some data seems unjustified (fecal volume, urine urea levels).
  • iIt is unclear how the nutritional targets were calculated in the early phase of admission - please explain.
  • Why did the authors feed the malnourished patients by the pareneteral route? 
  • The current guidelines do not support the prophylaxis of refeeding syndrome by parenteral nutrition - please explain.
  • The allocation of patients to pareneteral nutrition group raises most concerns on the validity of the study - the proposed inclusion criteria are not supported by the recent guidelines, and it is unclear why trophic feeding was not initiated at the earlier stage of treatment. What was the rationale for withdrawing enteral nutrition in the total parenteral group (gastric residual volume, abdominal distension, increased intrabdominal pressure)

Results

According to the results, the patients in parenteral and enteral nutrition groups are not statistically different. What was the rationale for allocating a significant number of patients to parenteral nutrition groups? There is a significant risk of selection bias; please explain.

Some of the statements are not scientifically valid and should be removed from the text (line 197-199)

The manuscript would benefit from removing some of the results, which are not significant. The tables are overloaded with data of little clinical significance. 

The conclusions are significant overstatements and require modification. 

Author Response

Reviewer # 2

The article describes feeding practices in critically ill patients admitted to an ICU due to a severe SARS CoV-2 infection. After careful examination, it is clear that the manuscript requires significant alternations before it is ready for publishing in the journal. The manuscript suffers from numerous flaws, including many significant issues such as:

General comments:

Both grammar and syntax must be improved - proper language editing is obligatory

We thank Reviewer#2 for the willingness and helpful comments. As suggested, all grammar and syntax corrections were made in the revised MS.

Introduction:

The authors failed to provide a convincing rationale for performing another study on the nutritional intervention in COVID-19. Furthermore, the introduction section is overloaded with insignificant data, primarily speculative. Please add information on the recently published guidelines (ESPEN, SSC, ASPEN) and rewrite the aims of the study, which are not clearly defined. 

 We thank Reviewer#2 for this helpful comment. As suggested, the introduction section was modified in the revised MS and the aims are now clearly defined.

Methodology

The manuscript would benefit from adding the STROBE checklist

The inclusion criteria are not described in sufficient detail - please use the STROBE checklist for guidance. 

 We thank Reviewer#2 for this helpful comment. As suggested, the STROBE checklist was added.

The feeding practices section requires significant modification:

According to the recent guidelines collection of some data seems unjustified (fecal volume, urine urea levels).

  • We thank Reviewer#2 for this helpful comment. As suggested, these data were deleted from the revised manuscript, but it is still a practice of our Unit to periodically measure data such as gastric residuals or fecal volume.

It is unclear how the nutritional targets were calculated in the early phase of admission - please explain.

We thank Reviewer#2 for this helpful comment. In the early phase of admission, caloric requirements were determined based on indirect calorimetry measurements in a subset of patients  or 70% of estimated target  for first week  using Penn State 2003 equation (see lines XXXX in the revised MS)

Why did the authors feed the malnourished patients by the pareneteral route?  The current guidelines do not support the prophylaxis of refeeding syndrome by parenteral nutrition - please explain.

  •  

We thank Reviewer#2 for this helpful comment. Intubation is associated to gastric distention and diaphragmatic function disorders (Tessler S, Kupfer Y, Lerman A, Arsura EL.Arch Intern Med. 1990 Feb;150(2):318-20), so we used parenteral feeding as a means of providing nutrition to malnourished patients, with high risk of reffeding and then the feeding route was chosen accordingly. Up to date, Three major societies also (ASPEN, IDA, INDI) recommend that PN is initiated as soon as possible in high nutrition risk and/or malnourished COVID 19 patients   (Martindale R., Patel J.J., Taylor B., Warren M., McClave S. American Society for Parenteral and Enteral Nutrition; May 2020. Nutrition therapy in the patient with COVID-19 disease requiring ICU care. Version 26., Irish Nutrition and Dietetic Institute . 2020. COVID-19 Dietetic care pathway Version 1 and guides to commencing enteral and parenteral nutrition in adult patients in intensive care with suspected or confirmed COVID-19 Version 2. Published March 2020. )

  • The allocation of patients to pareneteral nutrition group raises most concerns on the validity of the study - the proposed inclusion criteria are not supported by the recent guidelines, and it is unclear why trophic feeding was not initiated at the earlier stage of treatment. What was the rationale for withdrawing enteral nutrition in the total parenteral group (gastric residual volume, abdominal distension, increased intrabdominal pressure)

We thank Reviewer#2 for this helpful comment. As it was mentioned in previous comment, three major societies also (ASPEN, IDA, INDI) recommend that PN is initiated as soon as possible in high nutrition risk and/or malnourished patients. In the revised manuscript we have added now the following sentence: Parenteral nutrition was used in the presence of uncontrolled shock, uncontrolled life-threatening hypoxemia, hypercapnia or acidosis, hemodynamic instability or when energy intake was consistently <50% of targets over 5–7 days (see lines 175-178 in the revised MS)

Results

According to the results, the patients in parenteral and enteral nutrition groups are not statistically different. What was the rationale for allocating a significant number of patients to parenteral nutrition groups? There is a significant risk of selection bias; please explain.

We thank Reviewer#2 for this helpful comment. According to the results, the 2 groups differed only in Nutric Score values, side effects and time period of starting the feeding process. This reinforces our findings regarding potential effects of feeding modes on study final outcomes. Also, the allocation of patients to nutrition groups was guided based on the clinical picture and the possibility of using or not the intestinal tract.

Some of the statements are not scientifically valid and should be removed from the text (line 197-199)

The manuscript would benefit from removing some of the results, which are not significant. The tables are overloaded with data of little clinical significance. 

We thank Reviewer#2 for this helpful comment. We have now removed data of little clinical significance and statements on the text.

The conclusions are significant overstatements and require modification. 

We thank Reviewer#2 for this helpful comment. We have now modified the conclusion section accordingly.
